# Look at the Variance! Efficient Black-box Explanations with Sobol-based Sensitivity Analysis

**Thomas Fel**[1,3,4] [*]    **Rémi Cadène**[1,2] [*†]    **Mathieu Chalvidal**[1,3]
**Matthieu Cord**[2,5]    **David Vigouroux**[3,4]    **Thomas Serre**[1,3]

[1]Carney Institute for Brain Science, Brown University, USA  [2]Sorbonne Université, CNRS, France
[3]Artificial and Natural Intelligence Toulouse Institute, Université de Toulouse, France
[4] Institut de Recherche Technologique Saint-Exupery, France  [5]Valeo.ai
{thomas_fel,remi_cadene}@brown.edu

## Abstract

We describe a novel attribution method which is grounded in Sensitivity Analysis and uses Sobol indices. Beyond modeling the individual contributions of image regions, Sobol indices provide an efficient way to capture higher-order interactions between image regions and their contributions to a neural network's prediction through the lens of variance. We describe an approach that makes the computation of these indices efficient for high-dimensional problems by using perturbation masks coupled with efficient estimators to handle the high dimensionality of images. Importantly, we show that the proposed method leads to favorable scores on standard benchmarks for vision (and language models) while drastically reducing the computing time compared to other black-box methods – even surpassing the accuracy of state-of-the-art white-box methods which require access to internal representations. Our code is freely available: github.com/fel-thomas/Sobol-Attribution-Method.

## 1   Introduction

Deep neural networks are now being deployed in numerous domains including medicine, transportation, security or finances with broad societal implications. Yet, these networks have become nearly inscrutable, and for most real-world applications, these systems are used to make critical decisions – often without any explanation. In recent years, numerous explainability methods have been proposed [1, 2, 3, 4, 5, 6, 7, 8]. In addition to helping improve people's trust in these systems, these methods have helped identify and correct biases in datasets [4, 5, 9]. This has, in turn, helped improve these systems' robustness and accelerate their broad deployment. An important limitation of standard explainability methods is that they require access to the system's internal states including hidden layer activations or input gradients [1, 5, 6, 10]. As a result, these so-called white-box methods cannot be applied in the most general situations for which the internal states of network are not publicly accessible. For instance, it is common for companies to use neural networks provided by third parties (e.g., through web APIs or specialized hardware). However, only a handful of so-called black-box methods have been proposed to address this challenge with limited successes [2, 4, 8]. It is thus critical to develop more general methods that can reliably interpret and characterize the underlying decision processes of a wider array of models.

Common approaches to explaining a model's prediction consists of attributing a score for each input dimension such as image pixels for computer vision systems or individual words for natural language processing. Shown in Fig. 1 is an example image and associated importance map for an image categorization system, whereby scores for individual pixels are displayed as heatmaps where

---

[*] Equal contribution  [†]Work done before April 2021 and joining Tesla

35th Conference on Neural Information Processing Systems (NeurIPS 2021).

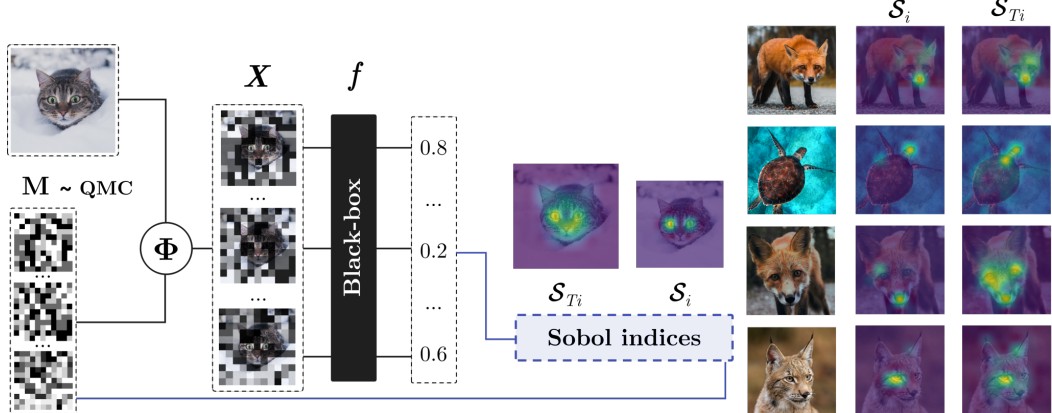

Figure 1: **(Left) Sobol Attribution Method overview.** Our method aims to explain the prediction of a black-box model for a given image. We first sample a set of real-valued masks $M$ drawn from a Quasi-Monte Carlo (QMC) sequence. We apply these masks to the input image through a perturbation function $\Phi$ (here the *Inpainting* function) to form perturbed inputs $X$ that we forward to the black box $f$ to obtain prediction scores. Using the masks $M$ and the associated prediction scores, we finally produce an explanation $S_{T_i}$ which characterizes the importance of each region by estimating the total order Sobol indices. While $S_{T_i}$ encompasses the effects of first and all higher-order non-linear interactions between pixel regions, we can also produce the first-order Sobol indices $S_i$ that reflect the importance of a region in isolation (e.g., the eyes of the cats). **(Right) Sample explanations for ResNet50V2.** Comparing explanations produced with $S_i$ and $S_{T_i}$ helps highlight the importance of individual image regions in isolation vs. jointly (e.g., the lynx tips are important but conditioned on the presence of the presence of an eye).

hotter locations correspond to pixels that contribute the most to the system's final prediction. In the context of black-box models, a core challenge is to derive these heatmaps using only the output predictions available through the network's forward pass. A simple approach consists in applying a given perturbation at a specific location on the input image to then measure how the corresponding prediction is affected. In the case of image models, pixel intensities are simply set to a default value corresponding to a pure black or gray value; in the case of language models, individual words are removed entirely from the text [11, 12]. However, evaluating the impact of these perturbations one dimension at a time fails to identify all of the non-linear interactions between input variables that are known to prevail in a complex system such as a deep neural network. However, estimating the combined effect of perturbations across multiple locations quickly becomes combinatorially intractable. Methods have been recently proposed to try to address some of these issues by grouping dimensions together, such as by grouping pixels within a neighborhood of the image (superpixel) [2] or sampling perturbation masks that affect multiple regions of the input [4, 8]. A first limitation of these approaches includes the use of Monte Carlo sampling methods which require a high number of forward passes – making these approaches computationally expensive. A second limitation is that they rely on relatively simple perturbations such as flipping pixels on or off [2, 4, 8]. This severely constrains the space of perturbations considered and limits the efficiency with which the space of perturbations can be explored.

We address these limitations by introducing an attribution method that leverages variance-based sensitivity analysis techniques, and more specifically Sobol indices [13]. These methods were initially introduced to help identify the input variables that have the most influence on the variance of the output of a non-linear mathematical system [14]. These were traditionally used in physics and economics to estimate the part of the variance induced by a group of variables on a system's output [15, 16, 17]. Our main contribution is a general framework to explain predictions from black-box models by adapting Sobol indices to be used in conjunction with perturbation masks. One of the originalities of our attribution method is that it is designed to support standard perturbations and real-valued intensity perturbations that can generate a continuous range of perturbations. Our second contribution is a tractable method for the calculation of Sobol indices. This is done by first sampling the perturbation masks following Quasi-Monte Carlo sequences, which efficiently covers the space of perturbations. This can be done best by leveraging the most efficient estimator borrowed from the sensitivity analysis literature [18, 19, 20, 21]. As a result, the proposed method can be

efficiently applied to high-dimensional inputs such as images. It produces on par with or better explanations than the state-of-the-art with at least half the number of forward passes. Another benefit of our method is that it allows to characterize not only the main effect of image regions but also higher-order interactions by decomposing the variance of the system's prediction using the Sobol indices. We run extensive experiments to demonstrate the benefits of the proposed method on several recent complementary benchmarks including the "Pointing Game" and "Deletion". Our primary focus is image classification, but our method is general and we include results on text classification benchmarks as well.

## 2 Related work

**Attribution methods for white-box models** A widely used family of attribution methods uses gradient computations via the backpropagation algorithm. The first such method was introduced in [22] and later refined in [1] in the context of deep convolutional networks. The idea is to backpropagate the gradient from a classification unit all the way down to the input pixels. The resulting gradient image indicates which pixels affect the decision score the most. Other gradient-based methods including DeConvNet [2] and Guided Backprop [3] were specifically developed to deal with certain activation functions such as ReLU. However, this family of methods are limited by the fact that they focus on the influence of individual pixels in an infinitesimal neighborhood around the input image in the image space. For instance, it has been shown that gradients often vanish when the prediction score to be explained is near the maximum value [7]. Integrated Gradient [7] partially addresses this issue by accumulating gradients along a straight interpolation path from a baseline state to the original image.

Yet another class of gradient-based methods [6, 23] iteratively optimize a perturbation mask over regions of an image. It leads to a different type of explanation given by the final perturbation mask. Such an approach might leverage more meaningful perturbations but it turns out to be computationally very slow with 20 seconds per explanation – limiting its broad applicability. In comparison, our proposed approach also works on image regions but it does not require iterative gradient computation – and can thus be parallelized.

Another family of attribution methods relies on the neural network's activations. Popular examples include CAM [24] which computes an attribution score based on a weighted sum of feature channels activities – right before the classification layer. GradCAM [5] extends CAM via the use of gradients to reweigh each feature channel to take into account their importance for the predicted class. These methods also work on image regions because they are computed on the output of feature maps which are themselves at a coarser resolution than the original image. In comparison, our proposed approach is model-agnostic and hence does not require access to internal computations.

**Attribution methods for black-box models** Most similar to our approach are attribution methods that can be used to explain the predictions of truly black-box models. These methods probe a neural network's responses to perturbations over image regions and combine the resulting predictions into an influence score for each individual pixel or group of pixels. The simplest method, "Occlusion" [2], masks individual image regions – one at a time – with an occluding mask set to a baseline value and assigns the corresponding prediction scores to all pixels within the occluded region. Then the explanation is given by these prediction scores and can be easily interpreted. However, occlusion fails to account for the joint (higher-order) interactions between multiple image regions. For instance, occluding two image regions – one at a time – may only decrease the model's prediction minimally (say a single eye or mouth component on a face) while occluding these two regions together may yield a substantial change in the model's prediction if these two regions interact non-linearly as is expected for a deep neural network.

Our work, together with related methods such as LIME [4] and RISE [8], addresses this problem by randomly perturbating the input image in multiple regions at a time. Obviously, perturbating multiple image locations simultaneously leads to a combinatorial explosion in the number of combinations and methods have been proposed to make these approaches more tractable. For instance, a popular method, LIME [4], takes superpixels as regions to perturbate instead of individual pixels. An influence score is then computed for a set of connected pixel patches indicating how strongly a patch is correlated to the model predictions.

RISE [8] relies on Monte Carlo sampling to generate a set of binary masks, each value in the masks representing a pixel region. By probing the model with randomly masked versions of the input,

RISE [8] produces a importance map by considering the average of the masks weighted by their associated prediction scores. Instead of using binary masks, our method considers a continuous range of perturbations which allows for a finer exploration of the model's response. Our method can still use the same perturbations as used in Occlusion [2], LIME [4] and RISE [8], but it also enables the use of more advanced perturbation functions that take continuous inputs.

More importantly, the aforementioned methods lack a rigorous framework. Here, we introduce a theoretical framework that decomposes the influence score of each individual region between multiple orders of influence. The first-order approximates Occlusion [2] by considering the influence of one region at a time, while the second-order considers two regions at a time, etc. The decomposition also includes higher-orders.

**Variance-based sensitivity analysis**    Our attribution method builds on the variance-based sensitivity analysis framework. The approach was introduced in the 70s [25] and reached a cornerstone with the Sobol indices [26]. Sobol indices are currently used in many fields (including those that are said to be safety-critical), especially for the analysis of physical phenomena [16]. They are used to identify the input dimensions that have the highest influence on the output of a model or a mathematical system. Several statistical estimators to compute these indices are available [15, 27, 20, 28, 29] and have asymptotic guarantees [20, 30, 31]. We build on this literature by adapting these Sobol indices in the context of black-box models to compute the influence of regions of an image on the output predictions using perturbation masks.

## 3    Sobol attribution method

In this work, we formulate the feature attribution problem as quantifying the contribution of a collection of $d$ real-valued features $\boldsymbol{x} = (x_1, ..., x_d)$ with respect to a model decision. Specifically, we consider a black-box decision function $\boldsymbol{f} : \mathcal{X} \to \mathbb{R}^k$ whose internal states and analytical form are unknown (for instance, $\boldsymbol{f}$ can score the probability for the input to belong to a specific class). Our goal is to quantify the importance of each feature to the decision score $\boldsymbol{f}(\boldsymbol{x})$, not just individually but also collectively. To capture these higher-order interactions, our method consists in estimating the Sobol indices of the features $\boldsymbol{x}$ by randomly perturbating them and evaluating the impact of these perturbations on the prediction of the black-box model (Fig. 1).

Considering variations of $\boldsymbol{f}(\boldsymbol{x})$ in response to meaningful perturbations of the input $\boldsymbol{x}$ is a natural way to interpret the local behavior of a the decision function $\boldsymbol{f}$ around $\boldsymbol{x}$. Several methods build on this idea, e.g., by removing one or a group of input features [2, 4, 6, 8, 23] or by back-propagating the gradient to the input space through the model [1, 7, 32, 5]. Most of these methods use the model's internal representations and/or require computing the gradient w.r.t. the input, which makes them unusable in a black-box setting. Moreover, these methods focus on estimating the intrinsic contribution of each feature, neglecting the combinatorial components. Our method applies perturbations directly on the input in order to deal with a black-box scenario, and allows us to estimate higher-order interactions between the features $\boldsymbol{x} = (x_1, ..., x_d)$ that contribute to $\boldsymbol{f}(\boldsymbol{x})$.

### 3.1    Random Perturbation

Formally, let us define a probability space $(\Omega, \mathcal{X}, P)$ of possible input perturbations and a random vector $\boldsymbol{X} = (X_1, ..., X_d)$ as a stochastic perturbation of the original input $\boldsymbol{x}$ (see Fig. 1). There are several ways to define random perturbations corresponding to different coverage of the data manifold around $\boldsymbol{x}$. For instance, we can consider the perturbation mask operator $\boldsymbol{\Phi} : \mathcal{X} \times \mathcal{M} \to \mathcal{X}$ which combines a stochastic mask $\boldsymbol{M} = (M_1, ..., M_d) \in \mathcal{M}$ (i.e., an i.i.d sequence of real-valued random variables on $[0, 1]^d$) with the original input $\boldsymbol{x}$. This formulation encompasses *Inpainting* perturbations: $\boldsymbol{\Phi}(\boldsymbol{x}, \boldsymbol{M}) = \boldsymbol{x} \odot \boldsymbol{M} + (\boldsymbol{1} - \boldsymbol{M})\mu$ with $\mu \in \mathbb{R}$ a baseline value, and $\odot$ the Hadamard product. This consists in linearly varying the pixel intensities towards a baseline intensity such as a pure black with a value of zero [6, 4, 2, 8]. Similarly, *Blurring* consists of applying a blur operator with various intensities to certain regions of the image [6]. Different perturbation domains can be considered for other types of data such as textual or tabular data that we discuss further in the experimental section. In the next section, we explain how we adapt the Sobol-based sensitivity analysis using a class of perturbations to explain the predictions of a black-box model.

### 3.2    Sensitivity analysis using Sobol indices

We first briefly review the classical Sobol-Hoeffding decomposition from [33] and introduce the Sobol indices. Let $(X_1, ..., X_d)$ be independent variables and assume that $\boldsymbol{f}$ belongs to $\mathbb{L}^2(\mathcal{X}, P)$.

Moreover we denote the set $\mathcal{U} = \{1, ..., d\}$, $\boldsymbol{u}$ a subset of $\mathcal{U}$, its complementary $\sim\!\mathbf{u}$ and $\mathbb{E}(\cdot)$ the expectation over the perturbation space. The Hoeffding decomposition allows us to express the function $\boldsymbol{f}$ into summands of increasing dimension, denoting $\boldsymbol{f_u}$ the partial contribution of variables $\boldsymbol{X}_u = (X_i)_{i \in \boldsymbol{u}}$ to the score $\boldsymbol{f}(\boldsymbol{X})$:

$$
\begin{aligned}
\boldsymbol{f}(\boldsymbol{X}) &= \boldsymbol{f}_\emptyset + \sum_i^d \boldsymbol{f}_i(X_i) + \sum_{1 \leqslant i < j \leqslant d} \boldsymbol{f}_{i,j}(X_i, X_j) + \cdots + \boldsymbol{f}_{1,...,d}(X_1, ..., X_d) \\
&= \sum_{\mathbf{u} \subseteq \mathcal{U}} \boldsymbol{f_u}(\boldsymbol{X_u})
\end{aligned}
\tag{1}
$$

Eq. 1 consists of $2^d$ terms and is unique under the following orthogonality constraint:

$$
\forall (\boldsymbol{u}, \boldsymbol{v}) \subseteq \mathcal{U}^2 \ s.t. \ \boldsymbol{u} \neq \boldsymbol{v}, \ \ \mathbb{E}(\boldsymbol{f_u}(\boldsymbol{X_u})\boldsymbol{f_v}(\boldsymbol{X_v})) = 0
\tag{2}
$$

Furthermore, orthogonality yields the characterization $\boldsymbol{f_u}(\boldsymbol{X}) = \mathbb{E}(\boldsymbol{f}(\boldsymbol{X})|\boldsymbol{X_u}) - \sum_{\boldsymbol{v} \subset \boldsymbol{u}} \boldsymbol{f_v}(\boldsymbol{X})$ and allows us to decompose the model variance as:

$$
\begin{aligned}
\mathrm{Var}(\boldsymbol{f}(\boldsymbol{X})) &= \sum_i^d \mathrm{Var}(\boldsymbol{f}_i(X_i)) + \sum_{1 \leqslant i < j \leqslant d} \mathrm{Var}(\boldsymbol{f}_{i,j}(X_i, X_j)) + ... + \mathrm{Var}(\boldsymbol{f}_{1,...,d}(X_1, ..., X_d)) \\
&= \sum_{\mathbf{u} \subseteq \mathcal{U}} \mathrm{Var}(\boldsymbol{f_u}(\boldsymbol{X_u}))
\end{aligned}
\tag{3}
$$

Building from Eq. 3, it is natural to characterize the influence of any input subset $\boldsymbol{u}$ as its own variance w.r.t. the total variance. This yields, after normalization by $\mathrm{Var}(\boldsymbol{f}(\boldsymbol{X}))$, the general definition of Sobol indices.

**Definition 1 (Sobol indices [26])** *The sensitivity index $\mathcal{S}_{\boldsymbol{u}}$ which measures the contribution of the variable set $\boldsymbol{X_u}$ to the model response $\boldsymbol{f}(\boldsymbol{X})$ in terms of fluctuation is given by:*

$$
\mathcal{S}_{\boldsymbol{u}} = \frac{\mathrm{Var}(\boldsymbol{f_u}(\boldsymbol{X_u}))}{\mathrm{Var}(\boldsymbol{f}(\boldsymbol{X}))} = \frac{\mathrm{Var}(\mathbb{E}(\boldsymbol{f}(\boldsymbol{X})|\boldsymbol{X_u})) - \sum_{\boldsymbol{v} \subset \boldsymbol{u}} \mathrm{Var}(\mathbb{E}(\boldsymbol{f}(\boldsymbol{X})|\boldsymbol{X_v}))}{\mathrm{Var}(\boldsymbol{f}(\boldsymbol{X}))}
\tag{4}
$$

Sobol indices give a quantification of the importance of any subset of features with respect to the model decision, in the form of a normalized measure of the model output deviation from $\boldsymbol{f}(\boldsymbol{X})$. Thus, Sobol indices sum to one : $\sum_{\boldsymbol{u} \subseteq \mathcal{U}} \mathcal{S}_{\boldsymbol{u}} = 1$.

For each subset of variables $\boldsymbol{X_u}$, the associated Sobol index $\mathcal{S}_{\boldsymbol{u}}$ describes the proportion of the model's output variance explained by this subset. In particular, the first-order Sobol indices $\mathcal{S}_i$ capture the intrinsic share of total variance explained by a particular variable, without taking into account its interactions. Many attribution methods construct such intrinsic importance estimator. However, the framework of Sobol indices enables us to capture higher-order interactions between features. In this view, we define the Total Sobol indices.

**Definition 2 (Total Sobol indices [34])** *The total Sobol index $\mathcal{S}_{T_i}$ which measures the contribution of the variable $X_i$ as well as its interactions of any order with any other input variables to the model output variance is given by:*

$$
\mathcal{S}_{T_i} = \sum_{\substack{\boldsymbol{u} \subseteq \mathcal{U} \\ i \in \boldsymbol{u}}} \mathcal{S}_{\boldsymbol{u}} = 1 - \frac{\mathrm{Var}_{\boldsymbol{X}_{\sim i}}(\mathbb{E}_{X_i}(\boldsymbol{f}(\boldsymbol{X})|\boldsymbol{X}_{\sim i}))}{\mathrm{Var}(\boldsymbol{f}(\boldsymbol{X}))} = \frac{\mathbb{E}_{\boldsymbol{X}_{\sim i}}(\mathrm{Var}_{X_i}(\boldsymbol{f}(\boldsymbol{X})|\boldsymbol{X}_{\sim i}))}{\mathrm{Var}(\boldsymbol{f}(\boldsymbol{X}))}
\tag{5}
$$

Where $\mathbb{E}_{\boldsymbol{X}_{\sim i}}(\mathrm{Var}_{X_i}(\boldsymbol{f}(\boldsymbol{X})|\boldsymbol{X}_{\sim i}))$ is the expected variance that would be left if all variables but $X_i$ were to be fixed. $\mathcal{S}_{T_i}$ is the sum of the Sobol indices for the all the possible groups of variables where $i$ appears, i.e. first and higher order interactions of variable $X_i$.

Since the total interaction index contains the first order index, it is natural that it is greater than or equal to the first order index. We thus note the property which can easily be deduced: $\forall i, 0 \leq \mathcal{S}_i \leq \mathcal{S}_{T_i} \leq 1$. We will now see why these two indices and the difference between them make them relevant for the explainability of a black-box model.

These statistics quantify the intrinsic (first-order indices) and relational (total indices) impact of each variable to the model output. A variable with a low total Sobol index is therefore not important to explain the model decision. Also, a variable has a weak interaction with other variables when $\mathcal{S}_{T_i} \approx \mathcal{S}_i$, while it has a strong interaction when the difference between its two indices is high. A strong interaction means that the effect of one variable on the variation of the model output depends on other variables. Thus, using Sobol indices allows to describe fine grained interactions between inputs which leads to the model decision. We next present an efficient method to estimate these indices.

## 3.3 Efficient estimator

As models are becoming more and more complex, the proposed estimator must take into account the computational cost of model evaluation. Many efficient estimators have been proposed in the literature [16]. In this work, we use the Jansen [19] estimator which is often considered as one of the most efficient [21]. Jansen is typically used with a Monte Carlo sampling strategy. We improve over Monte Carlo by using a Quasi-Monte Carlo (QMC) sampling strategy which generates low-discrepancy sample sequences allowing a faster and more stable convergence rate [35]. For more information, see appendix D. We will now describe the procedure to implement these estimators.

We start by drawing two independent matrices of size $N \times d$ of perturbation masks from a Sobol low discrepancy $LP_\tau$ sequences. $N$ will be our number of designs and we recall that $d$ is our dimensions (e.g, 121 for 11 by 11 masks). Once the perturbation operator is applied to $x$ with these masks, we obtain two matrices $\mathbf{A}$ and $\mathbf{B}$ of the same size as the perturbed inputs (i.e., partially masked images). We note $\mathbf{A}_{ji}$ and $\mathbf{B}_{ji}$ the elements of the matrices such that $i = 1, ..., d$ the number of variables studied and $j = 1, ..., N$ the number of samples in each matrix. We form the new matrix $\mathbf{C}^{(i)}$ in the same way as $\mathbf{A}$ except for the fact that the column corresponding to the variable $i$ is now replaced by the column of $\mathbf{B}$. We denote $f_\emptyset = \frac{1}{N} \sum_{j=0}^{N} \boldsymbol{f}(\mathbf{A}_j)$ and the empirical variance $\hat{\mathbf{V}} = \frac{1}{N-1} \sum_{j=0}^{N} (\boldsymbol{f}(\mathbf{A}_j) - f_\emptyset)^2$. The empirical estimators for first ($\hat{\mathcal{S}}_i$) and total order ($\hat{\mathcal{S}}_{T_i}$) can be formulated as:

$$\hat{\mathcal{S}}_i = \frac{\hat{\mathbf{V}} - \frac{1}{2N} \sum_{j=1}^{N} (\boldsymbol{f}(\mathbf{B}_j) - \boldsymbol{f}(\mathbf{C}_j^{(i)}))^2}{\hat{\mathbf{V}}} \qquad \hat{\mathcal{S}}_{T_i} = \frac{\frac{1}{2N} \sum_{j=1}^{N} (\boldsymbol{f}(\mathbf{A}_j) - \boldsymbol{f}(\mathbf{C}_j^{(i)}))^2}{\hat{\mathbf{V}}} \qquad (6)$$

Hence, to compute the set of first order and total indices, it is necessary to perform $N(d+2)$ forwards of the model. We study in section 3.3 how to choose a sufficient number of forwards ($N$). To ease understanding and demonstrate that these estimators can be easily implemented, we show in Algorithm 1 a minimal pythonic implementation of the total order estimator that outputs $\hat{\mathcal{S}}_{T_i}$ indices. The input Y contains the prediction scores of the $N * (d + 2)$ forwards. The scores are ordered following the same QMC sampling ordering of their associated mask. The output STis contains $d$ importance scores, one for each dimension of the mask. In the case of images, we obtain our final explanation map by applying a bilinear upsampling to match the dimensions of the input image.

```python
def total_order_estimator(Y, N=32, d=11*11):
  fA, fB, fC = Y[:N], Y[N:N*2], [Y[N*2+N*i:N*2+N*(i+1)] for i in range(d)]
  f0 = mean(fA)
  V = sum([(val - f0)**2 for val in fA]) / (len(fA) - 1)
  STis = [sum((fA - fC[i])**2) / (2 * N) / V for i in range(d)]
  return STis
```

Algorithm 1: Pythonic implementation of the Total Order indices ($\hat{\mathcal{S}}_{T_i}$) calculation.

### 3.3.1 Signed estimator

Although the proposed Sobol-based attribution method allows us to determine the impact of any variables for a given prediction and thus to identify diagnostic ones, it lacks the ability to highlight the type of contributions made, whether positive or negative. Simple methods such as "Occlusion" typically include this information. Hence, we propose a variant that combines the importance scores of the total Sobol indices with the sign of the occlusion. We compute the difference in score between the prediction on the original input $x$ and a partial version $x_{\setminus i}$ with the variable $x_i$ occluded. Intuitively, this provides an estimate of the direction of the variations generated by the variables studied with respect to a reference state.

$$\hat{\mathcal{S}}_{T_i}^{\Delta} = \hat{\mathcal{S}}_{T_i} \times \text{sign}(\boldsymbol{f}(x) - \boldsymbol{f}(x_{\setminus i})) \qquad (7)$$

| | Method | VOC07 Test (All/Diff) | | COCO14 Val (All/Diff) | | Time (s) |
|---|---|---|---|---|---|---|
| | | VGG16 | ResNet50 | VGG16 | ResNet50 | |
| | Baseline Center | 69.6 / 42.4 | 69.6 / 42.4 | 27.8 / 19.5 | 27.8 / 19.5 | - |
| White box | Saliency [1] | 76.3 / 56.9 | 72.3 / 56.8 | 37.7 / 31.4 | 35.0 / 29.4 | 0.031 |
| | DeconvNet [2] | 67.5 / 44.2 | 68.6 / 44.7 | 30.7 / 23.0 | 30.0 / 21.9 | 0.029 |
| | Guided-Backprop. [3] | 75.9 / 53.0 | 77.2 / 59.4 | 39.1 / 31.4 | 42.1 / 35.3 | 0.051 |
| | MWP [36] | 77.1 / 56.6 | 84.4 / 70.8 | 39.8 / 32.8 | 49.6 / 43.9 | 0.039 |
| | cMWP [36] | 79.9 / 66.5 | **90.7 / 82.1** | 49.7 / 44.3 | **58.5 / 53.6** | 0.040 |
| | GradCAM [5] | 86.6 / 74.0 | 90.4 / 82.3 | **54.2 / 49.0** | 57.3 / 52.3 | 0.015 |
| | ExtremalPerturbation [23] | **88.0 / 76.1** | 88.9 / 78.7 | 51.5 / 45.9 | 56.5 / 51.5 | 26.48 |
| Black box | RISE [8] | 86.9 / 75.1 | 86.4 / 78.8 | 50.8 / 45.3 | 54.7 / 50.0 | 13.19 |
| | Sobol ($\hat{\mathcal{S}}_{T_i}$) (ours) | 87.7 / 75.3 | **89.6 / 80.2** | 54.6 / 49.6 | 57.3 / 52.7 | 6.381 |
| | Sobol signed ($\hat{\mathcal{S}}_{T_i}^{\Delta}$) (ours) | **89.8 / 82.4** | 86.7 / 73.0 | **55.2 / 49.9** | **57.5 / 53.1** | 6.721 |

Table 1: **Pointing game.** Accuracy over the full test set and a subset of difficult images (defined in [36]). The first and second best results are **bolded** and underlined. Results are based on PyTorch re-implementations using the TorchRay package. The reported execution time is an average over 100 runs on ResNet50 using an Nvidia Tesla P100 on Google Colab and a batch size of 64. Lower execution time can be reached with higher batch size.

## 4 Experiments

To evaluate the benefits and the reliability of the Sobol attribution method, we performed multiple systematic experiments on vision and natural language models using common explainability metrics.

For our vision experiments, we compared the plausibility of the explanations produced on the Pointing Game [36] benchmark. We evaluate the fidelity of our explanations using the Deletion metric for 4 representative models commonly used in explainability studies: ResNet50V2 [37] , VGG16 [38], EfficientNet [39] and MobileNetV2 [40] trained on ILSVRC-2012 [41]. In addition, we also compared the speed of convergence of the proposed estimator with that of the leading approach, RISE [8], on the same models. For our NLP experiments, we fine-tuned a Bert model and trained a bi-LSTM on the IMDB sentiment analysis dataset [42] before comparing fidelity scores using word-deletion for representative methods.

Throughout this work, explanations were generated using the Sobol total estimator $\hat{\mathcal{S}}_{T_i}$ on the target class output. In the supplementary material, we demonstrate the effectiveness of modeling higher-order interactions between image regions by comparing $\hat{\mathcal{S}}_{T_i}$ against $\hat{\mathcal{S}}_i$ which only models the main effects. For the experiments involving images, the masks were generated at a resolution of $d' = 11 \times 11$ pixels, then upsampled with a nearest-neighbor interpolation method before being applied with the *Inpainting* perturbation function. Finally, $N$ was set to 32 which is equivalent to 3, 936 forward passes (see Section 3.3 for details). For $\hat{\mathcal{S}}_{T_i}^{\Delta}$, an occlusion using the same resolution as the masks was used to sign $\hat{\mathcal{S}}_{T_i}$, with zero as baseline. For RISE [8], we have followed the recommendations of the original paper with 8, 000 forward passes for all models.

### 4.1 Pointing game

Different evaluation methods have been proposed to compare attribution methods and their explanations [43, 44, 45, 46]. The first common approach consists in measuring the plausibility of an explanation as the correlation between attribution maps and human-provided semantic annotations. Here, we focused on the Pointing Game used in [36, 6, 23, 8]. For each attribution method, we compute a contribution score for each pixel of a given class of objects, e.g., bike or car. We then calculated the percentage of times the pixel with the highest score is included in the bounding box surrounding the object of interest. In this benchmark, a good attribution method should point to the most important evidence of the object appearance in accordance with a human user.

In Table 1, a report results for the Pascal VOC [47] and MS COCO [48] datasets using VGG16 [38] and ResNet50[37]. In the last column we report the computation times for each method averaged over 100 MS COCO samples for the ResNet50 model. We subdivided explanation methods into two

categories: white-box methods which require the use of backpropagation, such as Gradient [2] and Extremal Perturbation [23], and/or access to the internal states of the model, such as GradCAM [5] versus black-box methods such as Occlusion [2], RISE [8], or the proposed Sobol method $\mathcal{S}_{T_i}$ which only require the final model predictions. The proposed method outperforms RISE [8] on all of the tested cases, while reducing the number of forward passes by half. Surprisingly, white-box methods do not always lead to higher scores, and indeed $\hat{\mathcal{S}}_{T_i}^{\Delta}$ is the leading method for Pascal VOC / VGG16 and our two estimators $\hat{\mathcal{S}}_{T_i}^{\Delta}, \hat{\mathcal{S}}_{T_i}$ prevail on COCO / VGG16. Also note that our signed version of the estimator obtains higher scores overall. This might be due to the fact that images from VOC and COCO often feature several types of objects. Thus, the maximum variance in the output is not always induced by the object of interest but can be due to the masking of another object in the image. This result suggests that our signed version $\hat{\mathcal{S}}_{T_i}^{\Delta}$ should be used on multi-label datasets, while $\hat{\mathcal{S}}_{T_i}$ should be used on multi-class datasets. We indeed confirm this in the next set of experiments on a multi-class dataset.

## 4.2 Fidelity

There is a broad consensus that measuring the plausibility of an explanation alone is insufficient [49, 50]. Indeed, if an explanation is used to make a critical decision, users expect an explanation to reflect the true underlying decision process of the model and not just a consensus with humans. Failures to do so could have disastrous consequences. A first major limitation of current evaluation methods based on human-provided groundtruth such as the pointing game is that they do not work when a model prediction is wrong. In this case, an explanation method can be penalized for not pointing to the correct evidence even though explaining prediction errors is a critical use case for explanation methods. Another limitation of these evaluation methods is that they make the implicit assumption that the models should be relying on the same image regions than humans for recognition [51, 52, 53], which is likely to be an incorrect assumption. We thus use the fidelity metric as a complementary type of evaluation. This metric assumes that the more faithful an explanation is, the quicker the prediction score should drop when pixels that are considered important are reset to a baseline value (e.g., gray values).

In Table S1, we report results for the Deletion Metric [8] (or $1 - AOPC$ [43]) for 4 different pre-trained models: ResNet50 [37] , VGG16 [38], EfficientNet [39] and MobileNet [40] on 2,000 images sampled from the ImageNet validation set. TensorFlow [54] and the Keras [55] API were used to run the models. Several baseline values can be used [56], but we chose the standard approach with gray values. We observe that the proposed Sobol $\hat{S}{T_i}$ is the most faithful black-box methods with the lowest deletion scores across all models. Overall $\hat{\mathcal{S}}_{T_i}$ is able to match the scores of the most faithful white-box method, namely Integrated Gradients [7], and gets the lowest score on ResNet50V2 with 0.121 against 0.123 (lower is better). We also report that our signed version $\hat{\mathcal{S}}_{T_i}^{\Delta}$ is less faithful that the standard Sobol $\hat{S}{T_i}$. This can be explained by the fact that ImageNet images contains only one object and therefore the main variance area generally coincides with the class to be explained. This confirms our observation on the previous pointing game benchmark that $\hat{S}{T_i}$ should be preferred in a multi-class setup and $\hat{\mathcal{S}}_{T_i}^{\Delta}$ in a multi-label setup.

Another metric called Insertion has been proposed by the authors of RISE [8]. Instead of deleting pixels in the original image like with Deletion, Insertion consists in adding pixels on a baseline image, e.g. one gray image, starting with pixels that are associated with the highest importance scores for a given explanation method. An issue with Insertion is that the score computed along the insertion path is highly influenced by the first inserted pixels which contributes disproportionately. A good score on this metric therefore requires exploring a region very far from the original image and closer to the baseline. For this reason, we rather preferred to focus our study on Deletion than Insertion. However, we also report results on Insertion in the supplementary material using the same hyperparameters as used in Deletion.

## 4.3 Efficiency

The black-box methods presented so far compete with white-box methods that do not require access to the internal representation of the model at the cost of a large number of forward passes, e.g., around $8,000$ for RISE [8]. This weakness leads us to take a more serious look at the performance of the proposed method. It seems critical for the deployment of black-box methods to lower the amount of compute required to produce correct explanations. We describe an experiment to

| | Method | *ResNet50V2* | *VGG16* | *EfficientNet* | *MobileNetV2* |
|---|---|---|---|---|---|
| | Baseline Random (ours) | 0.235 | 0.168 | 0.124 | 0.137 |
| White box | Saliency [1] | 0.174 | 0.134 | 0.105 | 0.125 |
| | Guided-Backprop. [3] | 0.142 | 0.138 | 0.105 | 0.102 |
| | DeconvNet [2] | 0.159 | 0.146 | 0.105 | 0.111 |
| | Grad.-Input [57] | 0.140 | 0.096 | 0.093 | 0.103 |
| | Integ.-Grad. [7] | **0.123** | **0.095** | **0.091** | **0.093** |
| | SmoothGrad [32] | 0.130 | 0.106 | 0.094 | 0.098 |
| | GradCAM [5] | 0.141 | 0.118 | 0.130 | 0.122 |
| Black box | Occlusion [2] | 0.350 | 0.357 | 0.252 | 0.357 |
| | RISE [8] | 0.127 | 0.121 | 0.119 | 0.114 |
| | Sobol ($\hat{\mathcal{S}}_{T_i}$) (ours) | **0.121** | **0.109** | **0.104** | **0.107** |
| | Sobol signed ($\hat{\mathcal{S}}_{T_i}^{\Delta}$) (ours) | 0.145 | 0.114 | 0.147 | 0.141 |

Table 2: **Deletion** scores obtained on 2,000 ImageNet validation set images. Lower is better. Random consists in removing pixels at each step at random. The first and second best results are **bolded** and underlined.

show that beyond producing higher quality explanations, our estimator converges quickly. We first generate an explanation with a high number of forward passes that is large enough to reach convergence, e.g. $10,000$ forward passes. Then we compare this explanation that "converged" to other explanations obtained with lower numbers of forward passes. It allows us to measure the stability and rate of convergence towards this explanation that "converged", but more practically to find the proper trade-off between the amount of compute and the quality of explanations. This procedure requires defining a measure of similarity between two explanations. Since the proper interpretation method is to rank the features most sensitive to the model's decision, it seems natural to consider the Spearman rank correlation [58] to compare the similarity between explanations. Prior work has provided theoretical and experimental arguments in line with this choice [50, 49, 59, 46].

In Fig. 2, we compare the proposed Sobol attribution method $\hat{\mathcal{S}}_{T_i}$ against RISE [8], which is the current state-of-the-art for black-box methods. We use their respective gold explanation generated after $10,000$ forwards. To allow a fair comparison, both methods use masks generated in $7 \times 7$ dimensions (as recommended by RISE [8]). We report the average results and variance over $500$ images from the ImageNet validation set using Efficient-Net, a convolutional neural network optimized for fast forward computing times. We observe that our method exhibit higher convergence rate by getting higher Spearman's rank correlation of 0.8 after only 1,000 forwards against 0.65 for RISE, and consistently obtain higher scores until reaching 0.97 with 7,000 forwards against 0.73 for RISE. Additionally, we observe that Sobol has a more stable convergence by getting an overall lower

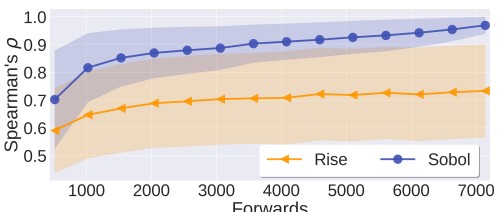

Figure 2: Comparison between the rate and stability of convergence of Sobol $\hat{\mathcal{S}}_{T_i}$ and RISE [8]. A high Spearman correlation rank corresponds to producing an explanation that is similar to the explanation that "converged", i.e. with $10,000$ forward passes. We report mean and variance computed over $500$ images from the ImageNet dataset using EfficientNet.

variance than RISE. This implies that the number of forward passes used in Sobol can be greatly reduced to accommodate computational resources constraints compared to RISE. Indeed, RISE proposes to use $8,000$ forwards, but our method is faster and reaches better results with half the number of passes. We report similar results for other neural network architectures in the supplementary material D. Finally, we perform an ablation study of Sobol to show the impact of lowering the number of forwards on the Deletion benchmark. We report competitive scores with 16 times fewer number of forwards than RISE by reaching 0.151 in Deletion score with 492 forwards. For reference, Sobol was reaching state-of-the-art results of 0.121 with 3,936 forwards. We report additional scores in the supplementary materials S2.

|       | Saliency | Grad-Input | SmoothGrad | Integ-Grad | Occlusion | $\hat{\mathcal{S}}_{T_i}$ | $\hat{\mathcal{S}}_{T_i}^{\Delta}$ |
|-------|----------|------------|------------|------------|-----------|--------------|---------------------|
| BERT  | 0.684    | 0.682      | 0.682      | 0.689      | **0.531** | 0.662        | 0.598        |
| LSTM  | 0.541    | 0.529      | 0.541      | 0.538      | **0.440** | 0.523        | 0.461        |

Table 3: **Word deletion** scores, obtained on 1,000 sentences. Delete up to 20 words per sentence accordingly to their relevance and track the impact on the classification performance. Lower is better. The first and second best results are **bolded** and underlined.

## 4.4 Word deletion

For NLP, black-box methods require the use of perturbations that can be applied to the space of characters, words or sentences. For instance, a common perturbation consists in simply removing one word of the sentence to be explained. Therefore, the *Inpainting* perturbation that we used with Sobol to reduce the intensity of pixels in a continuous manner cannot be directly applied in this context. Instead, we adapt it by binarizing the masks such that $\boldsymbol{\Phi}(\boldsymbol{x}, \mathbf{M}) = \boldsymbol{x} \odot \lceil \mathbf{M} - 0.5 \rceil$, i.e., if the value is greater than 0.5 the word is kept, otherwise it is removed. We then verify that our Sobol method can be used to identify words that support a specific decision of a text classifier. Inspired by previous work [11, 12, 60], we introduce an experimental benchmark on the IMDB Review dataset [61]. It is similar to the previous Deletion benchmark for images in that it focuses on assessing the faithfulness of the explanation and does not require specific human annotations. More precisely, we first trained two models: a bi-LSTM from scratch and a BERT model fine-tuned for the task (see appendix H for details). We generate explanations on 1,000 sentences from the validation dataset. An explanation associates an importance score to each word. Similar to Deletion, we use these scores to successively remove the most relevant word of the sentence and measure the corresponding drop in the prediction score of the model.

In Table 3, we report results for explanation methods that are commonly used in NLP. Both our two Sobol methods have a better faithfulness than all the tested gradient-based white-box methods including Saliency, Grad-Input, SmoothGrad, and Integr-Grad. For instance, Sobol $\hat{\mathcal{S}}_{T_i}^{\Delta}$ even reaches a low Word deletion scores of 0.461 (lower is better) for the bi-LSTM compared to 0.529 for the best white-box approach. However, the proposed methods are only the second and third most faithful methods. Occlusion (often called Omit-1 in NLP) reaches the lowest score of 0.440 for LSTM and 0.531 for BERT, against 0.461 and 0.598 for Sobol $\hat{\mathcal{S}}_{T_i}^{\Delta}$. This is due to the fact that the default distribution of masks used in Sobol is centered around 0.5, which corresponds to removing on average half of the words as opposed to a single word for Occlusion. In IMDB, this causes the frequent removal of critical words that support the model decision. Indeed, we report comparable results with Occlusion (e.g., 0.527 for BERT) for a lower threshold of 0.05 to remove far fewer words. Since Sobol can model higher-order interactions between words, we believe that it could successfully be used for NLP tasks that are more complex than sentiment classification on IMDB.

## 5 Conclusion

We have presented a novel explainability method to study and understand the predictions of a black-box model. This approach is grounded within the theoretical framework of sensitivity analysis using Sobol indices. A non-trivial contribution of this work was to make the approach tractable and efficient for high-dimensional data such as images. For this purpose, we have introduced a method using perturbation masks coupled with efficient estimators from the sensitivity analysis literature. One additional benefit of the approach is that it provides a way to study the importance of not just the main effects of input variables but also higher-order interactions between them. We showed that our method can be efficiently used to explain the decisions of image classifiers. It reaches performance on par with or better than the current best black-box methods while being twice as fast. It even reaches comparable results to the best white-box methods without requiring access to internal states. We also showed that our method could be applied to language models and reported initial competitive results. It is our hope that this work will guide further efforts in the search for more general and efficient explainability methods for black-box models and that further links will be made with the field of sensitivity analysis.

