## Acknowledgments

This work was conducted as part the DEEL* project. It was funded by the Artificial and Natural Intelligence Toulouse Institute (ANITI) grant #ANR19-PI3A-0004. MC was funded by ANR grant VISADEEP (ANR-20-CHIA-0022). TS was funded by ONR (N00014-19-1-2029) and NSF (IIS-1912280). The computing hardware was supported in part by NIH Office of the Director grant #S10OD025181 via the Center for Computation and Visualization.

## References

[1] Karen Simonyan, Andrea Vedaldi, and Andrew Zisserman. Deep inside convolutional networks: Visualising image classification models and saliency maps. In *Workshop Proceedings of the International Conference on Learning Representations (ICLR)*, 2014. 1, 3, 4, 7, 9, 19

[2] Matthew D Zeiler and Rob Fergus. Visualizing and understanding convolutional networks. In *Proceedings of the IEEE European Conference on Computer Vision (ECCV)*, 2014. 1, 2, 3, 4, 7, 8, 9, 15, 19

[3] Jost Tobias Springenberg, Alexey Dosovitskiy, Thomas Brox, and Martin Riedmiller. Striving for simplicity: The all convolutional net. In *Workshop Proceedings of the International Conference on Learning Representations (ICLR)*, 2014. 1, 3, 7, 9, 19

[4] Marco Tulio Ribeiro, Sameer Singh, and Carlos Guestrin. "why should i trust you?" explaining the predictions of any classifier. In *Proceedings of the 22nd ACM SIGKDD international conference on knowledge discovery and data mining*, 2016. 1, 2, 3, 4

[5] Ramprasaath R Selvaraju, Michael Cogswell, Abhishek Das, Ramakrishna Vedantam, Devi Parikh, and Dhruv Batra. Grad-cam: Visual explanations from deep networks via gradient-based localization. In *Proceedings of the IEEE International Conference on Computer Vision (ICCV)*, 2017. 1, 3, 4, 7, 8, 9, 19

[6] Ruth C Fong and Andrea Vedaldi. Interpretable explanations of black boxes by meaningful perturbation. In *Proceedings of the IEEE International Conference on Computer Vision (ICCV)*, 2017. 1, 3, 4, 7

[7] Mukund Sundararajan, Ankur Taly, and Qiqi Yan. Axiomatic attribution for deep networks. In *Proceedings of the International Conference on Machine Learning (ICML)*, 2017. 1, 3, 4, 8, 9, 19

[8] Vitali Petsiuk, Abir Das, and Kate Saenko. Rise: Randomized input sampling for explanation of black-box models. In *Proceedings of the British Machine Vision Conference (BMVC)*, 2018. 1, 2, 3, 4, 7, 8, 9, 15, 16, 19

[9] Corentin Dancette, Remi Cadene, Damien Teney, and Matthieu Cord. Beyond question-based biases: Assessing multimodal shortcut learning in visual question answering. In *Proceedings of the IEEE International Conference on Computer Vision (ICCV)*, 2021. 1

[10] Andrei Kapishnikov, Tolga Bolukbasi, Fernanda Viégas, and Michael Terry. Xrai: Better attributions through regions. In *Proceedings of the IEEE Conference on Computer Vision and Pattern Recognition (CVPR)*, 2019. 1

[11] Leila Arras, Franziska Horn, Grégoire Montavon, Klaus-Robert Müller, and Wojciech Samek. " what is relevant in a text document?": An interpretable machine learning approach. *PloS one*, 12(8):e0181142, 2017. 2, 10

[12] Leila Arras, Grégoire Montavon, Klaus-Robert Müller, and Wojciech Samek. Explaining recurrent neural network predictions in sentiment analysis. 2017. 2, 10

[13] I.M Sobol. Global sensitivity indices for nonlinear mathematical models and their monte carlo estimates. *Mathematics and Computers in Simulation*, 2001. 2, 15

[14] Andrea Saltelli. Making best use of model evaluations to compute sensitivity indices. *Computer physics communications*, 2002. 2

[15] Andrea Saltelli, Paola Annoni, Ivano Azzini, Francesca Campolongo, Marco Ratto, and Stefano Tarantola. Variance based sensitivity analysis of model output. design and estimator for the total sensitivity index. *Computer physics communications*. 2, 4

---

*https://www.deel.ai/

[16] Bertrand Iooss and Paul Lemaître. A review on global sensitivity analysis methods. *Uncertainty management in Simulation-Optimization of Complex Systems: Algorithms and Applications*, 2015. 2, 4, 6

[17] Thorsten Wagener and Francesca Pianosi. What has global sensitivity analysis ever done for us? a systematic review to support scientific advancement and to inform policy-making in earth system modelling. *Earth-science reviews*, 2019. 2

[18] Michiel JW Jansen, Walter AH Rossing, and Richard A Daamen. Monte carlo estimation of uncertainty contributions from several independent multivariate sources. In *Predictability and nonlinear modelling in natural sciences and economics*. Springer. 2

[19] Michiel J.W. Jansen. Analysis of variance designs for model output. *Computer Physics Communications*, 1999. 2, 6

[20] Alexandre Janon, Thierry Klein, Agnes Lagnoux, Maëlle Nodet, and Clémentine Prieur. Asymptotic normality and efficiency of two sobol index estimators. *ESAIM: Probability and Statistics*. 2, 4

[21] Arnald Puy, William Becker, Samuele Lo Piano, and Andrea Saltelli. A comprehensive comparison of total-order estimators for global sensitivity analysis. *International Journal for Uncertainty Quantification*, 2020. 2, 6

[22] David Baehrens, Timon Schroeter, Stefan Harmeling, Motoaki Kawanabe, Katja Hansen, and Klaus-Robert Müller. How to explain individual classification decisions. *The Journal of Machine Learning Research*, 11:1803–1831, 2010. 3

[23] Ruth Fong, Mandela Patrick, and Andrea Vedaldi. Understanding deep networks via extremal perturbations and smooth masks. In *Proceedings of the IEEE International Conference on Computer Vision (ICCV)*, 2019. 3, 4, 7, 8

[24] Bolei Zhou, Aditya Khosla, Agata Lapedriza, Aude Oliva, and Antonio Torralba. Learning deep features for discriminative localization. In *Proceedings of the IEEE Conference on Computer Vision and Pattern Recognition (CVPR)*, 2016. 3

[25] RI Cukier, CM Fortuin, Kurt E Shuler, AG Petschek, and JH Schaibly. Study of the sensitivity of coupled reaction systems to uncertainties in rate coefficients. i theory. *The Journal of chemical physics*, 59(8):3873–3878, 1973. 4

[26] Ilya M Sobol. Sensitivity analysis for non-linear mathematical models. *Mathematical modelling and computational experiment*, 1:407–414, 1993. 4, 5

[27] Amandine Marrel, Bertrand Iooss, Beatrice Laurent, and Olivier Roustant. Calculations of sobol indices for the gaussian process metamodel. *Reliability Engineering & System Safety*, 2009. 4

[28] Art B Owen. Better estimation of small sobol'sensitivity indices. *ACM Transactions on Modeling and Computer Simulation (TOMACS)*, 23(2):1–17, 2013. 4

[29] Stefano Tarantola, Debora Gatelli, and Thierry Alex Mara. Random balance designs for the estimation of first order global sensitivity indices. *Reliability Engineering & System Safety*, 2006. 4

[30] Sébastien Da Veiga and Fabrice Gamboa. Efficient estimation of sensitivity indices. *Journal of Nonparametric Statistics*, 2013. 4

[31] Jean-Yves Tissot and Clémentine Prieur. Bias correction for the estimation of sensitivity indices based on random balance designs. *Reliability Engineering & System Safety*, 2012. 4

[32] Daniel Smilkov, Nikhil Thorat, Been Kim, Fernanda Viégas, and Martin Wattenberg. Smoothgrad: removing noise by adding noise. In *Workshop on Visualization for Deep Learning, ICML*, 2017. 4, 9, 19

[33] Wassily Hoeffding. A class of statistics with asymptotically normal distribution. In *Annals of Mathematical Statistics*. 1948. 4

[34] Toshimitsu Homma and Andrea Saltelli. Importance measures in global sensitivity analysis of nonlinear models. *Reliability Engineering & System Safety*, 52(1):1–17, 1996. 5

[35] Mathieu Gerber. On integration methods based on scrambled nets of arbitrary size. *Journal of Complexity*, 2015. 6, 16

[36] Jianming Zhang, Sarah Adel Bargal, Zhe Lin, Jonathan Brandt, Xiaohui Shen, and Stan Sclaroff. Top-down neural attention by excitation backprop. In *Proceedings of the IEEE European Conference on Computer Vision (ECCV)*, 2016. 7

[37] Kaiming He, Xiangyu Zhang, Shaoqing Ren, and Jian Sun. Identity mappings in deep residual networks. In *Proceedings of the IEEE European Conference on Computer Vision (ECCV)*, 2016. 7, 8

[38] Karen Simonyan and Andrew Zisserman. Very deep convolutional networks for large-scale image recognition. 2015. 7, 8

[39] Mingxing Tan and Quoc Le. Efficientnet: Rethinking model scaling for convolutional neural networks. In *Proceedings of the International Conference on Machine Learning (ICML)*. PMLR, 2019. 7, 8

[40] Mark Sandler, Andrew Howard, Menglong Zhu, Andrey Zhmoginov, and Liang-Chieh Chen. Mobilenetv2: Inverted residuals and linear bottlenecks. In *Proceedings of the IEEE Conference on Computer Vision and Pattern Recognition (CVPR)*, 2018. 7, 8

[41] J. Deng, W. Dong, R. Socher, L.-J. Li, K. Li, and L. Fei-Fei. ImageNet: A Large-Scale Hierarchical Image Database. In *Proceedings of the IEEE Conference on Computer Vision and Pattern Recognition (CVPR)*, 2009. 7

[42] Andrew L. Maas, Raymond E. Daly, Peter T. Pham, Dan Huang, Andrew Y. Ng, and Christopher Potts. Learning word vectors for sentiment analysis. In *Proceedings of the 49th Annual Meeting of the Association for Computational Linguistics: Human Language Technologies*. 7

[43] Wojciech Samek, Alexander Binder, Grégoire Montavon, Sebastian Lapuschkin, and Klaus-Robert Müller. Evaluating the visualization of what a deep neural network has learned. *IEEE transactions on neural networks and learning systems*, 2016. 7, 8

[44] Sara Hooker, Dumitru Erhan, Pieter-Jan Kindermans, and Been Kim. A benchmark for interpretability methods in deep neural networks. In *Advances in Neural Information Processing Systems (NIPS)*, 2019. 7

[45] Umang Bhatt, Adrian Weller, and José M. F. Moura. Evaluating and aggregating feature-based model explanations. In *Proceedings of the International Joint Conference on Artificial Intelligence (IJCAI)*, 2020. 7

[46] Thomas Fel and David Vigouroux. Representativity and consistency measures for deep neural network explanations. *Proceedings of the IEEE Winter Conference on Applications of Computer Vision (WACV)*, 2022. 7, 9

[47] Mark Everingham, Luc Van Gool, Christopher KI Williams, John Winn, and Andrew Zisserman. The pascal visual object classes (voc) challenge. Springer, 2010. 7

[48] Tsung-Yi Lin, Michael Maire, Serge Belongie, James Hays, Pietro Perona, Deva Ramanan, Piotr Dollár, and C Lawrence Zitnick. Microsoft coco: Common objects in context. In *Proceedings of the IEEE European Conference on Computer Vision (ECCV)*. Springer, 2014. 7

[49] Julius Adebayo, Justin Gilmer, Michael Muelly, Ian Goodfellow, Moritz Hardt, and Been Kim. Sanity checks for saliency maps. *Advances in Neural Information Processing Systems (NeurIPS)*, 2018. 8, 9, 15, 19

[50] Amirata Ghorbani, Abubakar Abid, and James Zou. Interpretation of neural networks is fragile. In *Proceedings of the AAAI Conference on Artificial Intelligence (AAAI)*, 2017. 8, 9, 15

[51] Shimon Ullman, Liav Assif, Ethan Fetaya, and Daniel Harari. Atoms of recognition in human and computer vision. *Proceedings of the National Academy of Sciences*, 2016. 8

[52] Drew Linsley, Sven Eberhardt, Tarun Sharma, Pankaj Gupta, and Thomas Serre. What are the visual features underlying human versus machine vision? In *Proceedings of the IEEE International Conference on Computer Vision Workshops*, 2017. 8

[53] Drew Linsley, Dan Shiebler, Sven Eberhardt, and Thomas Serre. Learning what and where to attend. *Proceedings of the International Conference on Learning Representations (ICLR)*, 2018. 8

[54] Martín Abadi, Ashish Agarwal, Paul Barham, Eugene Brevdo, Zhifeng Chen, Craig Citro, Greg S. Corrado, Andy Davis, Jeffrey Dean, Matthieu Devin, Sanjay Ghemawat, Ian Goodfellow, Andrew Harp, Geoffrey Irving, Michael Isard, Yangqing Jia, Rafal Jozefowicz, Lukasz Kaiser, Manjunath Kudlur, Josh Levenberg, Dandelion Mané, Rajat Monga, Sherry Moore, Derek Murray, Chris Olah, Mike Schuster, Jonathon Shlens, Benoit Steiner, Ilya Sutskever, Kunal Talwar, Paul Tucker, Vincent Vanhoucke, Vijay Vasudevan, Fernanda

Viégas, Oriol Vinyals, Pete Warden, Martin Wattenberg, Martin Wicke, Yuan Yu, and Xiaoqiang Zheng. TensorFlow: Large-scale machine learning on heterogeneous systems, 2015. Software available from tensorflow.org. 8

[55] François Chollet et al. Keras. https://keras.io, 2015. 8

[56] Pascal Sturmfels, Scott Lundberg, and Su-In Lee. Visualizing the impact of feature attribution baselines. 2020. 8

[57] Avanti Shrikumar, Peyton Greenside, Anna Shcherbina, and Anshul Kundaje. Not just a black box: Learning important features through propagating activation differences. *arXiv preprint arXiv:1605.01713*, 2016. 9, 17, 19

[58] Charles Spearman. The proof and measurement of association between two things. *American Journal of Psychology*, 1904. 9

[59] Richard Tomsett, Dan Harborne, Supriyo Chakraborty, Prudhvi Gurram, and Alun Preece. Sanity checks for saliency metrics. In *Proceedings of the AAAI Conference on Artificial Intelligence (AAAI)*, 2019. 9

[60] Sebastian Bach, Alexander Binder, Grégoire Montavon, Frederick Klauschen, Klaus-Robert Müller, and Wojciech Samek. On pixel-wise explanations for non-linear classifier decisions by layer-wise relevance propagation. *PloS one*, 10(7):e0130140, 2015. 10

[61] Andrew L. Maas, Raymond E. Daly, Peter T. Pham, Dan Huang, Andrew Y. Ng, and Christopher Potts. Learning word vectors for sentiment analysis. In *Proceedings of the 49th Annual Meeting of the Association for Computational Linguistics: Human Language Technologies*, 2011. 10

[62] Dylan Slack, Sophie Hilgard, Emily Jia, Sameer Singh, and Himabindu Lakkaraju. Fooling lime and shap: Adversarial attacks on post hoc explanation methods. In *Proceedings of the AAAI/ACM Conference on AI, Ethics, and Society*, pages 180–186, 2020. 15

[63] Il'ya Meerovich Sobol'. On the distribution of points in a cube and the approximate evaluation of integrals. *USSR Computational Mathematics and Mathematical Physics*, 1967. 16

[64] Gunther Leobacher and Friedrich Pillichshammer. *Introduction to quasi-Monte Carlo integration and applications*. Springer, 2014. 16

[65] Marco Ancona, Enea Ceolini, Cengiz Öztireli, and Markus Gross. Towards better understanding of gradient-based attribution methods for deep neural networks. In *Proceedings of the International Conference on Learning Representations (ICLR)*, 2018. 17

[66] Avanti Shrikumar, Peyton Greenside, and Anshul Kundaje. Learning important features through propagating activation differences. In *Proceedings of the International Conference on Machine Learning (ICML)*, 2017. 17

[67] Matthew Sotoudeh and Aditya V. Thakur. Computing linear restrictions of neural networks. In *Advances in Neural Information Processing Systems (NIPS)*, 2019. 18

[68] Christian Szegedy, Vincent Vanhoucke, Sergey Ioffe, Jon Shlens, and Zbigniew Wojna. Rethinking the inception architecture for computer vision. In *Proceedings of the IEEE Conference on Computer Vision and Pattern Recognition (CVPR)*, 2016. 19, 20

[69] Sepp Hochreiter and Jürgen Schmidhuber. Long short-term memory. *Neural computation*, 1997. 20

[70] Jeffrey Pennington, Richard Socher, and Christopher D Manning. Glove: Global vectors for word representation. In *Proceedings of the Conference on Empirical Methods in Natural Language Processing (EMNLP)*, 2014. 20

[71] Thomas Wolf, Julien Chaumond, Lysandre Debut, Victor Sanh, Clement Delangue, Anthony Moi, Pierric Cistac, Morgan Funtowicz, Joe Davison, Sam Shleifer, et al. Transformers: State-of-the-art natural language processing. In *Proceedings of the 2020 Conference on Empirical Methods in Natural Language Processing: System Demonstrations*, 2020. 20

[72] Diederik P Kingma and Jimmy Ba. Adam: A method for stochastic optimization. *Proceedings of the International Conference on Learning Representations (ICLR)*, 2014. 20

# A  Broader Impact

This work is part of the burgeoning field of explainable AI which has a potential positive impact on society including transportation, science and medicine. There is now broad consensus that many (if not all) the AI systems deployed in real-world settings exhibit significant biases including gender and racial biases. Understanding how these systems arrive at their decisions is a necessary first step before these biases can be corrected. We described a method that provides explanations for predictions made by black-box models that are as faithful as possible (in the sense that it reflects the true inner-working of the model). We thus feel it is important for any explainability method to be built on a rigorous theoretical framework. Here, we borrowed methods from Sensitivity Analysis which has been extensively used to evaluate critical systems and that we showed compare favorably to other approaches on fidelity benchmarks. Nevertheless, progress in explicability should not result in a blind trust in the explanations of the models, which should always be used in the knowledge of their associated flaws. Finally, the attribution methods presented in this work are sensitive to adversarial attacks that can be used to hide the behavior of a model [62, 50] which is still an open problem in the frame of attribution methods.

# B  Qualitative comparison

Regarding the visual consistency of our method, Fig. S1 shows a side-by-side comparison between our method and the other methods tested in the Fidelity benchmark. The images are not hand-picked but are the first images from the ImageNet validation set. To allow better visualization, the gradient-based methods were 2 percentile clipped. The only black box methods are Occlusion, Rise and $\mathcal{S}_{T_i}$. We found that $\mathcal{S}_{T_i}$ consistently provides a sparser map than RISE [8] while being equally consistent. On the other hand, we found that in general, the gradient-based method provides the sharpest map, but some are prone to failure (fourth row in the Fig. S1), which is a known problem [49].

# C  Effectiveness of modeling higher-order interactions

We introduced two approaches, Sobol ($\hat{\mathcal{S}}_{T_i}$) and Sobol signed ($\hat{\mathcal{S}}_{T_i}^{\Delta}$), that combine effects of first- and all higher-orders interactions between image regions. For comparison, Occlusion [2] only accounts for the first order as it removes one region at a time, while RISE [8] accounts for higher-order by removing around 50% of regions at a time. As seen in Table S1, RISE already surpasses Occlusion on ImageNet in term of Deletion scores, which may indicate that using higher-order information is effective.

To further demonstrates that it is critical to model the higher orders, we evaluate Sobol first-order ($\mathcal{S}_i$) on our Deletion benchmark. We report that Sobol ($\mathcal{S}_{T_i}$) reaches lower deletions scores (lower is better) than Sobol first-order ($\mathcal{S}_i$) with 0.121 against 0.170 respectively on ResNet50v2, and similar differences on VGG16, EfficientNet and MobileNetV2.

| Method | ResNet50V2 | VGG16 | EfficientNet | MobileNetV2 |
|---|---|---|---|---|
| Sobol first-order ($\hat{\mathcal{S}}_i$) | 0.170 | 0.147 | 0.129 | 0.143 |
| Sobol ($\hat{\mathcal{S}}_{T_i}$) | **0.121** | **0.109** | **0.104** | **0.107** |

Table S1: **Deletion** scores obtained on 2,000 ImageNet validation set images. Lower is better.

# D  Efficiency of Sobol estimator

Regarding the estimation of the Sobol indices, we notice that we can derive a 'brute-force' (or often called double-loop method [13]) estimator from the definition 2:

$$\mathcal{S}_i = \frac{\int [\int \boldsymbol{f}(\boldsymbol{X}) d\boldsymbol{X}_{\sim i}]^2 d\boldsymbol{X}_i - (\int \boldsymbol{f}(\boldsymbol{X}) d\boldsymbol{X})^2}{\int \boldsymbol{f}(\boldsymbol{X})^2 d\boldsymbol{X} - (\int \boldsymbol{f}(\boldsymbol{X}) d\boldsymbol{X})^2} \tag{8}$$

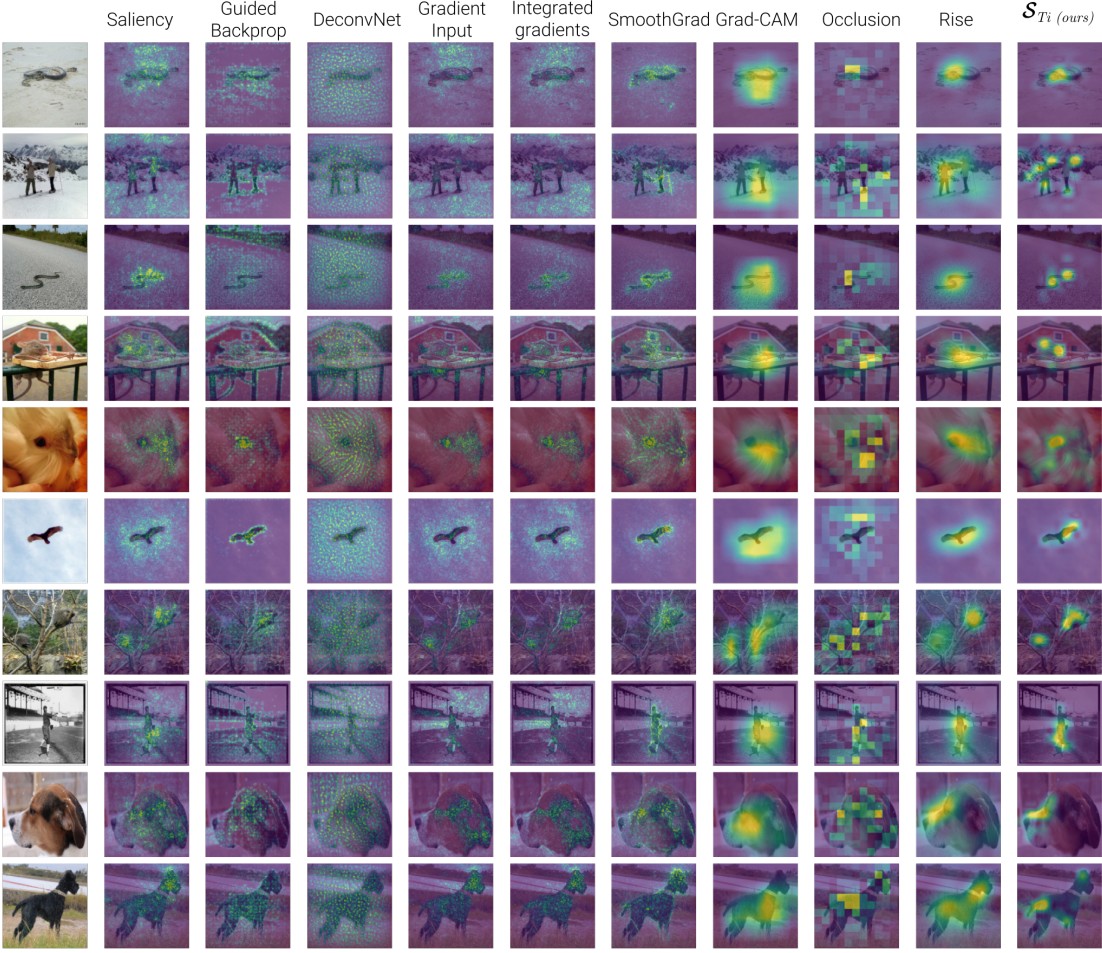

Figure S1: **Qualitative comparison** with other explainability methods. The heatmaps are normalized and clipped at 2 percentile for Saliency, Guided-Backprop, DeconvNet, Smoothgrad and Integrated-Gradients. Explanations are generated from a ResNet50V2.

However, one the main problems with this estimator is the cost of computation, which can be too heavy, especially with complex models such as large neural networks. This difficulty is particularly true for the calculation of total Sobol indices.

Since the perturbation masks are used to approximate these integrals, an efficient way to proceed is to generate those masks from a low discrepancy sequences, also called Quasi-random sequences. These sequences allow to efficiently integrate functions on the hypercube $[0, 1]^d$. In fact, they have a faster convergence rate compared to ordinary Monte Carlo methods [35] (with $f$ sufficiently regular). This difference being due to the use of a deterministic sequence that covers $[0, 1]^d$ more uniformly. In our experiments we used Sobol sequences [63], we refer the readers to [64] for more informations. The efficiency of the estimator and the sampling is shown on Figures S2, S3 and S4 where our estimator consistently converges faster than RISE [8].

We also perform an ablation study of the number of forwards on the Deletion benchmark. In Table S2, we show that competitive scores can be obtained with lower number of forwards such as 0.151 in Deletion score with 492 forwards instead of 0.121 with 3936 forwards which is our default number of forwards.

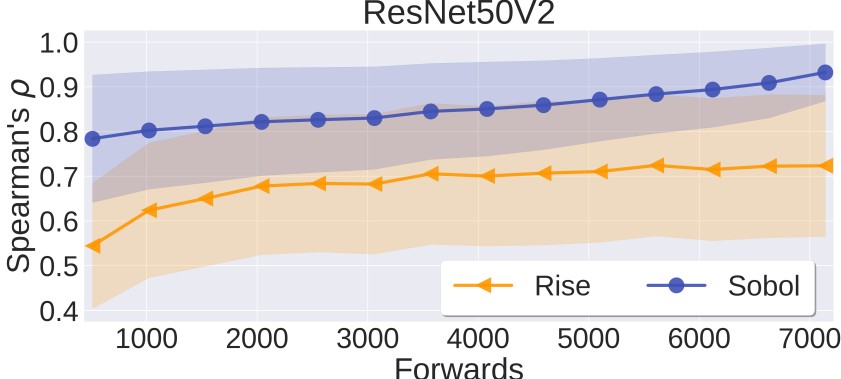

Figure S2: Spearman rank correlation of explanations as a function of the number of forwards, compared to an explanation generated with $10,000$ forwards. The model used is a ResNet50V2.

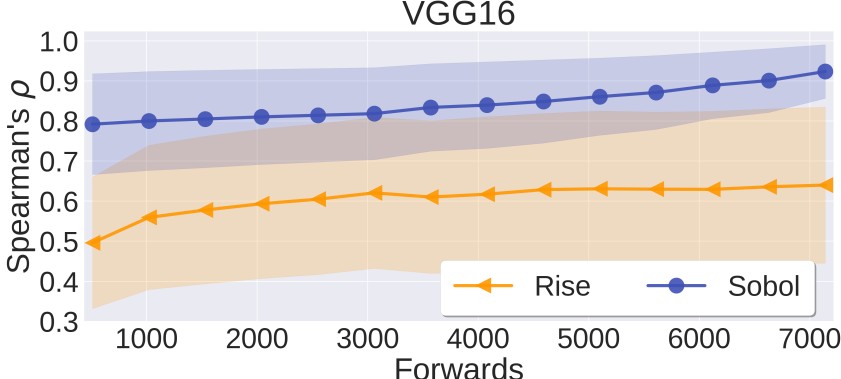

Figure S3: Spearman rank correlation of explanations as a function of the number of forwards, compared to an explanation generated with $1,0000$ forwards. The model used is a VGG16.

## E    Explanation methods

In the following section, the formulation of the different methods used in the experiment is given. We define $\boldsymbol{f}(\boldsymbol{x})$ the logit score (before softmax) for the class of interest. An explanation method provides an attribution score for each input variables. Each value then corresponds to the importance of this feature for the model results.

**Saliency** is a visualization techniques based on the gradient of a class score relative to the input, indicating in an infinitesimal neighborhood, which pixels must be modified to most affect the score of the class of interest.

$$g^{SA}(\boldsymbol{x}) = ||\nabla_{\boldsymbol{x}} \boldsymbol{f}(\boldsymbol{x})||$$

**Gradient $\odot$ Input** [57] is based on the gradient of a class score relative to the input, element-wise with the input, it was introduced to improve the sharpness of the attribution maps. A theoretical analysis conducted by [65] showed that Gradient $\odot$ Input is equivalent to $\epsilon$-LRP and DeepLIFT [66] methods under certain conditions: using a baseline of zero, and with all biases to zero.

$$g^{GI}(\boldsymbol{x}) = \boldsymbol{x} \odot ||\nabla_{\boldsymbol{x}} \boldsymbol{f}(\boldsymbol{x})||$$

**Integrated Gradients** consists of summing the gradient values along the path from a baseline state to the current value. The baseline is defined by the user and often chosen to be zero. This integral can be approximated with a set of $m$ points at regular intervals between the baseline and the point

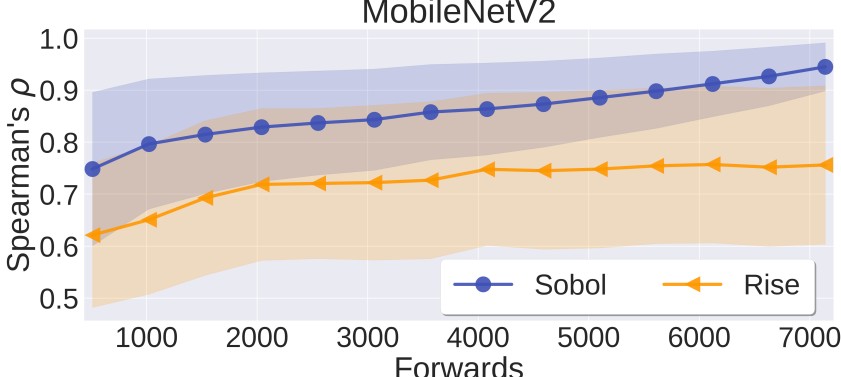

Figure S4: Spearman rank correlation of explanations as a function of the number of forwards, compared to an explanation generated with $1,0000$ forwards. The model used is a MobileNetV2.

| Number of samples | Deletion scores |
|---|---|
| 492 | 0.151 |
| 984 | 0.140 |
| 1476 | 0.132 |
| 1968 | 0.123 |
| 2460 | 0.121 |
| 2952 | 0.120 |
| 3444 | 0.120 |
| 3936 | 0.121 |

Table S2: **Deletion** scores averaged over 2,000 images of ImageNet validation set using ResNet50V2 and Sobol ($\hat{\mathcal{S}}_{T_i}$). Lower is better.

of interest. In order to approximate from a finite number of steps, we use a Trapezoidal rule and not a left-Riemann summation, which allows for more accurate results and improved performance (see [67] for a comparison). The final result depends on both the choice of the baseline $\mathbf{x}_0$ and the number of points to estimate the integral. In the context of these experiments, we use zero as the baseline and $m = 80$.

$$g^{IG}(\boldsymbol{x}) = (\boldsymbol{x} - \boldsymbol{x}_0) \int_0^1 \nabla_{\boldsymbol{x}} \boldsymbol{f}(\boldsymbol{x}_0 + \alpha(\boldsymbol{x} - \boldsymbol{x}_0)))d\alpha$$

**SmoothGrad** is also a gradient-based explanation method, which, as the name suggests, averages the gradient at several points corresponding to small perturbations (drawn i.i.d from a normal distribution of standard deviation $\sigma$) around the point of interest. The smoothing effect induced by the average help reducing the visual noise, and hence improve the explanations. In practice, Smoothgrad is obtained by averaging after sampling $m$ points. In the context of these experiments, we took $m = 80$ and $\sigma = 0.2$ as suggested in the original paper.

$$g^{SG}(\boldsymbol{x}) = \mathbb{E}_{\varepsilon \sim \mathcal{N}(0, \boldsymbol{I}\sigma)}(\nabla_{\boldsymbol{x}} \boldsymbol{f}(\boldsymbol{x} + \varepsilon))$$

**Grad-CAM** can be used on Convolutional Neural Network (CNN), it uses the gradient and the feature maps $\mathbf{A}^k$ of the last convolution layer. More precisely, to obtain the localization map for a class, we need to compute the weights $\alpha_c^k$ associated to each of the feature map activation $\mathbf{A}^k$, with $k$ the number of filters and $Z$ the number of features in each feature map, with $\alpha_k^c = \frac{1}{Z} \sum_i \sum_j \frac{\partial \boldsymbol{f}(\boldsymbol{x})}{\partial \mathbf{A}_{ij}^k}$ and

$$g^{GC} = max(0, \sum_k \alpha_k^c \mathbf{A}^k)$$

Notice that the size of the explanation depends on the size (width, height) of the last feature map, a bilinear interpolation is performed in order to find the same dimensions as the input.

**Occlusion** is a sensitivity method that sweep a patch that occludes pixels over the images, and use the variations of the model prediction to deduce critical areas. In the context of these experiments, we took a patch size and a patch stride of 20.

$$g_i^{OC} = \boldsymbol{f}(\boldsymbol{x}) - \boldsymbol{f}(\boldsymbol{x}_{[\boldsymbol{x}_i=0]})$$

**RISE** is a black-box method that consist of probing the model with randomly masked versions of the input image to deduce the importance of each pixel using the corresponding outputs. The masks $\boldsymbol{m} \sim \mathcal{M}$ are generated randomly in a subspace of the input space, then upsampled with a bilinear interpolation (once upsampled the masks are no longer binary).

As recommended in the original paper, we used $N = 8,000$ and $\mathbb{E}(\mathcal{M}) = 0.5$ for all the experiments.

$$g^{RI}(\boldsymbol{x}) = \frac{1}{\mathbb{E}(\mathcal{M})N} \sum_{i=0}^{N} \boldsymbol{f}(\boldsymbol{x} \odot \boldsymbol{m}_i)\boldsymbol{m}_i$$

## F  Fidelity with Insertion

Insertion is an evaluation procedure introduced in [8] at the same time as Deletion. Deletion assumes that the more faithful an explanation is, the faster the prediction score should drop when pixels that are considered important are reset to a baseline value (e.g., gray values). Insertion is the opposite of Deletion in that it assumes that the prediction score should go up faster for the most faithful explanations when pixels from the original image that are considered important are added to a baseline image (e.g., gray image). Metrics similar to Insertion are less common than Deletion in the literature that is why we focus on Deletion in the main paper. Moreover, just like Deletion, the Insertion score is largely influenced by the first steps [8]: the first pixels removed from the original image for deletion, and the first pixels inserted in the insertion case. Maximizing Insertion means exploring in a space close to the baseline, while maximizing Deletion means exploring a space around the original image. We suggest that for this reason, the Insertion score is not as relevant as Deletion.

| | Method | *ResNet50V2* | *VGG16* | *EfficientNet* | *MobileNetV2* |
|---|---|---|---|---|---|
| | Random Baseline (ours) | 0.233 | 0.166 | 0.115 | 0.138 |
| White box | Saliency [1] | 0.363 | 0.303 | 0.229 | 0.253 |
| | Guided-Backprop. [3] | 0.377 | 0.242 | 0.229 | 0.361 |
| | DeconvNet [2] | 0.307 | 0.221 | 0.229 | 0.166 |
| | Grad.-Input [57] | 0.194 | 0.219 | 0.098 | 0.126 |
| | Integ.-Grad. [7] | 0.264 | 0.237 | 0.143 | 0.166 |
| | SmoothGrad [32] | 0.445 | 0.374 | 0.299 | 0.307 |
| | GradCAM [5] | **0.524** | **0.438** | **0.393** | **0.419** |
| Black box | Occlusion [2] | 0.154 | 0.115 | 0.152 | 0.135 |
| | RISE [8] | **0.546** | **0.484** | **0.439** | **0.443** |
| | Sobol ($\hat{\mathcal{S}}_{T_i}$) (ours) | 0.370 | 0.313 | 0.309 | 0.331 |
| | Sobol signed ($\hat{\mathcal{S}}_{T_i}^{\Delta}$) (ours) | 0.258 | 0.290 | 0.204 | 0.211 |

Table S3: **Insertion** scores, obtained on 2,000 images from ImageNet validation set. Higher is better. Random consists in inserting randomly among the pixels remaining at each step. The first and second best results are **bolded** and underlined.

## G  Sanity check

We followed the procedure used by [49], namely the progressive reset of the network weights. We used an Inception V3 [68] model, each images shows the $\mathcal{S}_{T_i}$ explanation for the network in which

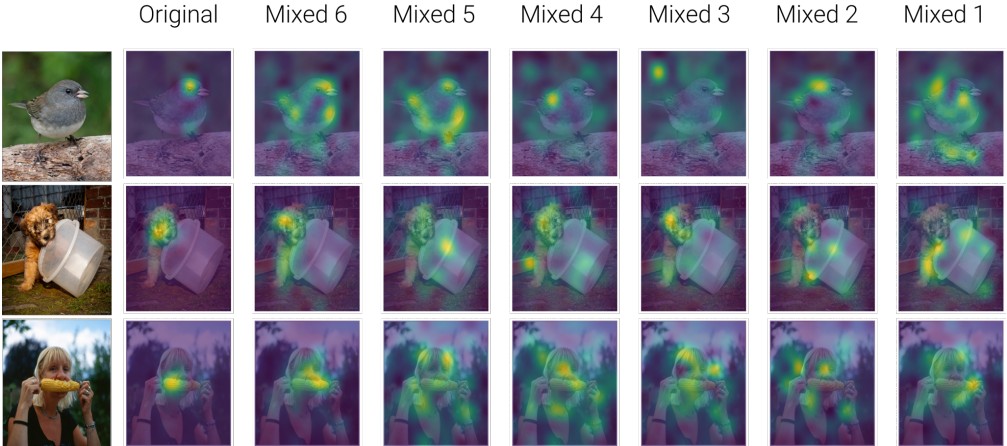

Figure S5: **Sanity Check** model weights are progressively reinitialized from Mixed 6 to Mixed 1 in InceptionV3 [68], demonstrating our method's sensitivity to model weights.

the upper layers (from logits) were reset. Fig. S5 shows that our method passes the sanity check: it turns out to be sensitive to the modification of the model weights.

## H   Word Deletion

For the bidirectional LSTM [69], the word embedding is in $\mathbb{R}^{300}$ and is initialized with the pre-trained GloVe embedding [70]. The layer has a hidden size of $64$ (bidirectional architectures: $32$ dimensions per direction). The resulting document representation is projected to $64$ dimensions then $2$ dimensions using fully connected layers, followed by a softmax and reached an accuracy of $89\%$ on the test dataset.

For the BERT-based models, we use the Transformers library from HuggingFace [71] and more specifically the bert-base-uncased model. The final layer is tuned to minimize cross-entropy, with Adam optimizer [72] and initial learning rate of $1e^{-3}$ to reach an accuracy of $92\%$ on the test dataset.

The observation that local perturbation: with the majority of words present, gets a better score is verified by playing on the threshold of the perturbation function. By decreasing the percentage of words removed on average we observe that a better deletion score is obtained.

|  | $\hat{\mathcal{S}}_{T_i}\Delta\ 50\%$ | $\hat{\mathcal{S}}_{T_i}^{\Delta}\ 90\%$ | $\hat{\mathcal{S}}_{T_i}\Delta\ 95\%$ | Occlusion |
|---|---|---|---|---|
| Deletion | 0.598 | 0.553 | **0.527** | 0.531 |

Table S4: **Word deletion scores** on the Bert based model when the perturbation threshold is modified to control the average presence of words in each generated perturbated input. Lower is better.