# OpenReview forum: "Look at the Variance! Efficient Black-box Explanations with Sobol-based Sensitivity Analysis"
_NeurIPS.cc/2021/Conference — NeurIPS 2021 Poster_

### Official Review · Reviewer_Szkm · 2021-07-14

**Rating:** 6
**Confidence:** 2

**Summary:**

This paper proposes a new method to understand the input-output causal relationship of a deep network. The proposed method is based on input perturbation (black-box), Quasi-Monte Carlo is used to sample two real-value masks for each input. Sobol indices is applied to measure the variance discrepancy between all the feature and a subset of the total. The sampling method for perturbation is efficient, and the sobol indices can capture both first- and high-order information of the variance. The proposed method can better visualise and understand the causal relationship of a learned network.

**Limitations And Societal Impact:**

I personally like the clarity of this draft, the aim of this study is stated clearly in Section 1(a black-box method for causal relationship analysis), and the method is explained in details in Section 3 (sobol indices can deliver high-order information and Quasi_monte carlo can reduce the sampling complexity). And to my knowledge, this study of high-order information is important to better understand which regions(based on their interaction) are used for making a prediction. (other reviewers can complete the literature I missed.) If the rebuttal can explain the question I raised in the last comment, I would like to accept this study.


**Main Review:**

I have seen some interpretability studies proposed based on variance or entropy, but they can only show first-order information. The proposed method can capture high-order information. From Fig 1, the high-order information could be different from the first-order info. Given the receptive field of a model is bigger than the input in most cases, the high-order information could play a more significant role in decision making. Thus, this high-order information study is more intuitive and reasonable to me.

One question in my mind is about the real-value mask. The use of binary mask is intuitive that some of the features(or input) will be completely ignored. When applying a real-value mask, does this actually create new patterns, e.g., a specific pixel or region could have low values due to the mask. Is it safe to consider the low-real-value region is somehow "masked out" for the variance computation, or could the weight will be increased to compensate this mask? I might miss some background for this setup, so I hope I can see some explanations in the rebuttal. If this is not a common setting, it is better to discuss a little in the paper. (L111 did not answer this question.)

Generally, the idea of applying sobol indices is interesting to me.

**Time Spent Reviewing:**

three

---

> ### Author Response · Authors · 2021-08-11
> **Answer to Reviewer Szkm**
>
> We are pleased that you found our paper interesting and liked its clarity.
>
> Q: “When applying a real-value mask, does this actually create new patterns, e.g., a specific pixel or region could have low values due to the mask. Is it safe to consider the low-real-value region is somehow "masked out" for the variance computation, or could the weight will be increased to compensate for this mask? I might miss some background for this setup, so I hope I can see some explanations in the rebuttal. If this is not a common setting, it is better to discuss a little in the paper”
>
> 1. Applying a real-value mask creates new patterns as shown in Figure 1: regions of the perturbed inputs X have different intensities. Intuitively, real-valued intensities allow to more precisely probe the model’s response by varying the contrast of a region instead of just masking-out that region.
> 2. One should not assume that low intensity regions are simply “masked-out” but one can assume that the contributions of these regions in the output prediction is reduced.  Identifying which regions are more sensitive is at the core of the proposed method.
> 3. Using real-value masks is new, but a similar setup can be found in simpler forms in the white-box literature: Integrated Gradient [9] or XRAI [11] applies different reduction methods of the global intensity of the image (all pixels at a time) before calculating gradients, and SmoothGrad [27] applies random noise patterns to vary the pixel intensities. We will extend this discussion, especially around L111, and highlight the differences with binary masks used by previous works (Occlusion and RISE).

---

### Official Review · Reviewer_n1GW · 2021-07-16

**Rating:** 7
**Confidence:** 4

**Summary:**

This paper proposes a black-box method to assess the contribution of image pixels and groups of pixels (regions) to the classification decision. The proposed approach is based on Sobol indices, that allow to characterize the influence of an input subset (image pixel/group of pixels or region) as its own variance with respect to the total variance.
To employ these indices in an efficient way, they are approximated by empirical Sobol indices. Specifically, from an input image, a set of real-valued masks is drawn from a Quasi-Monte-Carlo Sequence to generate perturbed inputs. The Sobol indices and total Sobol indices are estimated empirically from the variance of model predictions of these images. The first-order Sobol indices indicate the importance of each pixel in isolation, the total Sobol  indices also encompass higher-order interactions.
The method is evaluated on standard benchmarks for vision and NLP and shows competitive performance to black box and white box methods.

**Limitations And Societal Impact:**

yes

**Main Review:**

The paper proposes a new method to asses the contribution of image pixels and regions to the classification score of an image, of the contribution of words in the case of NLP applications. The proposed paper/approach has several strengths:

- The motivation to consider the relative variance of the prediction is clear and interesting.
- The combination with the occlusion approach yields practically good results and allows to assess positive vs. negative impact to a score.
- The paper is well written and mathematically clear.
- The experimental evaluation for vision and language applications is thorough.
- The efficiency is also properly evaluated

Weaknesses and Questions:
- A discussion on the discrepancy between the approximated, empirical variance and the actual variance would be interesting. Also, the accuracy of this approximation depends on the number of samples drawn. An ablation study on the needed number of samples would be interesting.
- considering the result in the pointing game, the a version of the proposed approach outperformes RISE in every case. Yet, the signed Sobol variant is not always the best.
- in the NLP evaluation, the Occlusion method works better than the proposed approach. I think this needs more discussion. For example, what is the role of masking during training in this context?

Overall, the paper proposes an efficient model that can be useful to the community and that is well motivated.

**Time Spent Reviewing:**

3h

---

> ### Author Response · Authors · 2021-08-11
> **Answer to Reviewer n1GW**
>
> We are glad that you found our paper useful and well motivated, and we are thankful for your suggestions.
>
> Q: “[Could you add] a discussion on the discrepancy between the approximated, empirical variance and the actual variance? Also, the accuracy of this approximation depends on the number of samples drawn.  An ablation study on the needed number of samples would be interesting.”
>
> 1. The discrepancy between the approximated and actual variance depends on the estimator. We use the Jansen estimator (see L199). It has been proven to efficiently converge to the actual variance with a probability of error that decreases with the number of samples. We will add this precision and cite Theorem 4 of  [B]. *[B] Global sensitivity indices for nonlinear mathematical models and their Monte Carlo estimates, I.M. Sobol'*
> 2. The accuracy of the approximation indeed depends on the number of samples drawn as studied in Figure 2 (and supp. material). We can see that after around 3000 samples adding more samples does not significantly improve the explanation (spearman correlation > 0.9). That is why we chose 3936 samples (see L236) which decreases the execution time of our estimator by 2 compared to RISE. We will clarify this point in the final version.
> 3. As suggested, we will add the following ablation study of the number of samples
>
> | Number of samples | Deletion scores (lower is better) |
> |---|---|
> | 492   | 0.151 |
> | 984   | 0.140 |
> | 1476 | 0.132 |
> | 1968 | 0.123 |
> | 2460 | 0.121 |
> | 2952 | 0.120 |
> | 3444 | 0.120 |
> | 3936 | 0.121 |
>
> Q: “In the NLP evaluation, the Occlusion method works better than the proposed approach. I think this needs more discussion. For example, what is the role of masking during training in this context?“
>
> 1. We hypothesize that Occlusion works best on IMDB because for this task only a few words are usually important to reach correct predictions such as “good”, “bad”, “not” and Occlusion removes one word at a time. In contrast, our method can remove varying numbers of words in the sentence. We report in L341, that our method can reach similar results to Occlusion (0.527 with BERT) when we set the threshold to remove less words (0.05%). We will extend the discussion in Section 4.4.
> 2. We are not sure what you mean by “the role of masking during training in this context”. As said in L324, the Inpainting perturbation with real-value masks is not applicable in NLP, so we binarize the masks. Please advise if needed.

---

> > ### Comment · Reviewer_n1GW · 2021-08-19
> > **Discussion and ablation**
> >
> > Thanks for the discussion and the additional ablation study. These should be reflected in a final version of the paper.

---

### Official Review · Reviewer_Xxm2 · 2021-07-17

**Rating:** 4
**Confidence:** 4

**Summary:**

In this work, the authors propose to use Sobol indices for black-box explanations of deep neural networks on computer vision and NLP tasks. The method is based on efficiently estimating Sobol indices and Total Sobol indices with a Quasi-Monte Carlo sampling strategy. The experiments on the tasks of pointing game, deletion, and word deletion show that the proposed method achieves comparable results to state-of-the-art attribution models.


**Limitations And Societal Impact:**

The limitation is not clearly explained yet. The potential negative societal impact may include wrong explanation to deep learning models.

**Main Review:**

Strengths:
+ The proposed approach is based on the Sobol indices that have been used for sensitivity analysis in many fields.
+ The authors propose to use the Quasi-Monte Carlo sampling strategy for efficient estimation of Sobol indices
+ The baselines and experiments are comprehensive.

Weaknesses:
- The technical novelty appears to be limited. The proposed method is the adoption of existing techniques.
- The experimental results do not show that the method has outstanding performance, according to Table 1,2,3.
- I do not see why black-box attribution models are necessary for practice applications. Can authors say some applications with black-box attribution models?
- The experiments have not demonstrated that high-order information is effective in the proposed model.


**Time Spent Reviewing:**

2

---

> ### Author Response · Authors · 2021-08-11
> **Answer to Reviewer Xxm2**
>
> Thank you for your review. We are happy to address your comments. The paper will improve as a result.
>
> Q: “I do not see why black-box attribution models are necessary for practice applications. Can authors say some applications with black-box attribution models?”
>
> 1. Black-box methods are already fairly well established and our work builds on prior work. For instance, a recently accepted paper [A] (oral CVPR 2021) uses RISE [10] (oral BMVC 2018) which is SOTA for black-box.
>       *[A] Black-box explanation of object detectors via saliency maps, Petsiuk et al. 2021*
> 2. RISE has shown that it is possible to outperform white box methods without needing to access the internal state of the model, with the only limitation that RISE is more computationally heavy. However, ours achieves better or comparable results to RISE while requiring 2x less computation (as can be seen in Table 1, 2 and Figure 2).
> 3. As suggested, we will add specific applications where only methods like ours or RISE can be used. It is very common for companies to use neural networks provided by third parties (e.g., through web APIs or specialized hardware which do not provide access to internal states or allow to compute gradients) and hence black-box methods become critical for those use cases.
>
> Q: “[Could you demonstrate] that high-order information is effective in the proposed model?”
>
> 1. Our approach characterizes the combined effects of first- and all higher-orders interactions between image regions. For comparison, Occlusion only accounts for the first order as it removes one region at a time, while RISE accounts for higher-order as it removes around 50% of regions at a time. As seen in Table 2, RISE already surpasses Occlusion on ImageNet which demonstrates that using higher-order information is effective.
> 2. As suggested, we will add the evaluation of the first order separately. Here are some additional Deletion scores to the current Table 2. We can see that using total order allows to reach lower deletion scores (lower is better) than using the first order only. These results further demonstrate that the higher-order information is effective in the proposed model.
>
> | Method | Resnet50V2 | VGG16 | EfficientNet | MobileNetV2 |
> |---|---|---|---|---|
> | Sobol first order only (new) | 0.170 | 0.147 | 0.129 | 0.143 |
> | Sobol total order | 0.121 | 0.115 | 0.104 | 0.107 |

---

> > ### Author Response · Authors · 2021-08-26
> > **Update results**
> >
> > Dear reviewer,
> >
> > As promised, we updated our table of results to address your remark on the effectiveness of the higher-order information.
> >
> > Best regards

---

### Decision · Program_Chairs · 2021-09-27

**Decision:**

Accept (Poster)

**Comment:**

A method for assessing the influence of image regions on neural network predictions is proposed.  Reviewers n1GW and Szkm are positive about the approach and its reported experimental performance.  In discussion following the author response, Reviewer Xxm2 stated the response's ablation study sufficiently addressed concerns.